# Assessing newborn scoring with each resuscitation (ANSWER): Protocol for identifying and testing an Apgar score for the 21st century

Henry J. Rozycki[1]*, Elizabeth E. Foglia[2], Miheret S. Yitayew[1], Heidi M. Herrick[2]

1 Division of Neonatal Medicine, Children's Hospital of Richmond at VCU and Department of Pediatrics, School of Medicine, Virginia Commonwealth University, Richmond, Virginia, United States of America,
2 Division of Neonatology, Department of Pediatrics, The Children's Hospital of Philadelphia, Perelman School of Medicine at University of Pennsylvania, Philadelphia, Pennsylvania, United States of America

* hrozycki@vcu.edu

## Abstract

### Background

The status of infants in the minutes after birth has been summarized by the Apgar score for the last 70 years and is applied at most medically attended deliveries around the world. It has not, however, adapted to changes in neonatal resuscitation over the decades. There are issues with application to premature newborns, how to account for the interventions outlined in the Newborn Resuscitation Program (NRP), inter-rater reliability, and local custom. Developing a modern newborn assessment score will require a series of steps, the first of which is to identify which observations or data can best differentiate a normal newborn transition from an abnormal one.

### Methods

Video recordings of at least 35 random normal (meeting NRP goals for heart rate and saturation without intervention, normal physical exam and requiring only normal postnatal care) and 35 random abnormal (not meeting at least 2 normal criteria) from 7 centers participating in the VERIFI study. The presence or absence of observable elements, including vital signs, appearance, and interventions will be recorded every 30 seconds for the first 5 minutes of life. Observations, as well as any changes over time or after intervention will be compared between normal and abnormal, and those that are significant and independent based on logistic regression will become candidate newborn assessment score components.

### Expected results

There are likely to be ten or more observations/elements from videos of the first five minutes of life that will differ between normal and abnormal newborns, and these will

**Data availability statement:** No datasets were generated or analysed during the current study. All relevant data from this study will be made available upon study completion.

**Funding:** The project described was supported by the National Center for Advancing Translational Sciences (NCATS), National Institutes of Health, through Grant Award Number 5R21TR003994 (HJR and EEF). The funders had no role in study design, data collection and analysis, decision to publish, or preparation of the manuscript.

**Competing interests:** The authors have declared that no competing interests exist.

be tested in all combinations to identify the 1–3 score sets that will be applied to a new set of VERIFI videos to identify which has the best sensitivity/specificity.

## Conclusions

Bringing newborn assessment into modern practice will build on the legacy of Dr. Apgar. A series of steps, beginning with identifying the observations/elements that best identify newborns who may need further care can lead to a universal validated tool for the 21st century.

---

## Introduction

The status of almost every baby born in the US, Europe and most other countries is assessed over the first five minutes of life using the system that Virginia Apgar published almost 70 years ago [1]. Along with the date of birth, it is the first number assigned to millions each year. The initial purpose, according to Dr. Apgar, was to identify newborns with difficult transitions to extrauterine life who might therefore require further support, but it quickly evolved to become a risk factor/predictor for mortality [2] and for near [3]- and long-term [4] morbidities for the baby. Over the next decades, as it became almost universal, it also became an outcome marker for problems during pregnancy and delivery. In a review of 500 papers published in 2018–19, the Apgar score was used to assess issues and interventions in pregnancy (35%), mode of delivery and maternal anesthesia (30%), as a risk factor for short and long-term problems for the newborn (19%), and as a descriptor of the newborn's condition (16%) [5].

The Collaborative Perinatal Project included the Apgar score among the prospective collected data on over 50,000 pregnancies from 1959–1965 and demonstrated an association between low scores and both mortality [2] and one-year neurodevelopmental problems [4]. Obstetrical care, newborn care, and newborn resuscitation have markedly changed since then, but no contemporary studies have examined the impact of these changes on the utility of the Apgar score. There are also major concerns regarding its prognostic accuracy [6], application to premature newborns [7], inter-rater reliability [8,9], how to account for ongoing resuscitative interventions [10], and applicability across different countries and regions [11]. Accordingly, there are increasing calls to overhaul newborn scoring to make it more relevant or even to abolish it [12–14].

Given how a simple observational scoring system for all newborns is almost universally applied at delivery and used by so many medical (and non-medical) specialties, a newborn assessment methodology for the 21st century must be 1) simple to apply, 2) informed by easily obtained data, primarily from observation at the assessment time, 3) adaptable to any circumstance at delivery, and 4) consistent across observers and local practices.

Revising the approach to newborn assessment will require a series of steps, including reassessment of relevant inputs, feasibility testing, and real-world

comparisons with the current gold standard, the Apgar score. The first step is to determine which potential components of an assessment tool best differentiate a normal and successful transition to extrauterine life from one that is not normal and/or identifies the newborns who require increased medical attention at delivery or after postnatal transition. The Assessing Newborns With Each Resuscitation (ANSWER) study is designed to create a hierarchy of candidate newborn assessment components using video recordings of newborn resuscitation.

## Design

The data for this study will be collected from analysis of video recordings as part of the VERIFI (Video Evaluation for Resuscitation of Infants International) project, an international registry of data captured from video recorded delivery room resuscitations to characterize practice, process and outcomes of delivery room resuscitation. The goal is to collect 300–450 videos per year and follow the subjects until hospital discharge or transfer. Currently, seven centers are preparing to participate: the Hospital of the University of Pennsylvania, Oklahoma Children's Hospital OU Health, Stavanger University Hospital, Universitätskliniken der MedUni Wien, St Louis University/Cardinal Glennon Children's Hospital, Technisches University Dresden, and Universiteit Leiden. At each center, video- and audio-recordings of the newborn are captured in the first minutes of life. The videos are stored within each site per their individual policies and only anonymized data derived from analysis of the videos done at each site are uploaded to the registry. The ANSWER study will leverage the VERIFI infrastructure to assess the feasibility and inter-rater reliability of candidate measures of newborn health that can be ascertained from a convenience sample of these video recordings. Two raters at each site will independently review and score each video for each of the candidate elements listed in Table 1 and to classify each delivery outcome as Normal, Abnormal, or Indeterminate. Time to first cry and breath will be recorded at the exact time. The other elements will be recorded at every 30 seconds after birth for the first 5 minutes. Observations of the newborn such as heart rate, saturation, and elements derived from the Apgar score will be recorded as close to but preceding each 30-second mark. Intervention elements will be scored if they occurred in the thirty second time preceding the assessment rime. For example, if CPAP is applied at 75 seconds of life, and removed at 185 seconds of life, it would recorded as Yes/Present only at 90, 120, 150 and 180 seconds. If an element cannot be accurately assessed at any of the successive 30 second marks, it will be recorded as Unobservable.

Components will be dichotomized (yes/no). In the case of continuous variables, this will be done for a series of cut-offs (e.g., for heart rate, 0, < 50 bpm, < 60 bpm, < 80 bpm, < 100 bpm). For graded variables, each grade will be scored as yes/no (e.g., for Apgar score tone, which has 3 grades, there will be 3 potential yes/no components). An additional set of components will be derived from changes in a particular measure or observation and the time that the change took (e.g., a subject with heart rate <60 at 90 secs, < 90 at 120 secs and > 100 at 150 secs will be coded as rise/inadequate 120, rise/adequate 150) to see if recovery can help differentiate normal from abnormal transition. These improvements will also be separately coded as spontaneous or after intervention.

Clinical data accompanying each video will include the estimated gestational age at delivery, sex, mode of delivery, and birthweight rounded to 250g increments. Videos will be categorized into Normal (all 5 criteria met), Abnormal (≤ 2 criteria met) or Indeterminant (3–4 criteria met) using the checklist below. Only normal and abnormal videos will be used for this study:

a) preductal oxygen saturation measurements meeting NRP goals without supplementation, if measured

b) Heart rate > 100 throughout postnatal transition.

c) no apnea or respiratory distress.

d) no resuscitation interventions.

e) normal postnatal care. Note that for preterm infants, this will be defined as discharge at ≤ 40 weeks estimated gestational age, with a normal neurological exam, in room air, and normal oral breast or formula feedings, and without

**Table 1. Candidate scoring elements from resuscitation videos.**

| Observation/Element | Definition |
|---|---|
| Time to first breath | Record exact time (e.g., 17 seconds) |
| Time to first cry | Record exact time (e.g., 17 seconds) |
| **Elements assessed every 30 seconds of life to 5 minutes** | |
| MEASURES | |
| O2 saturation | Record actual number at or closest to but prior to each 30 second epoch. |
| FiO2 | Maximum during the preceding 30 second epoch. |
| Heart Rate | Record actual number at or closest to but prior to each 30 second epoch. Record method heart rate was ascertained (e.g., auscultation, pulse oximeter, ECG) |
| NEWBORN STATUS | |
| Color | Assessed at each 30 second mark: Blue or Pale – face/body Acrocyanotic – hands and/or feet Pink – face/body/extremities |
| Respirations | Apnea/No chest movement Gasping/severe retractions Inadequate/small/irregular Adequate/Normal/Crying |
| Tone | Limp/Flaccid Some flexion/reduced for gestational age Active motion/appropriate for gestational age |
| Neuro | No reflex response/staring Incomplete or reduced for gestational age response Normal/appropriate for gestational age response |
| Activity [21] | Absent Flexed arms/legs Active |
| Grimace [21] | No response Minimal response Prompt response |
| INTERVENTIONS | |
| CPAP | Y/N administered during the preceding 30 second epoch |
| Oxygen Delivery | Y/N FiO2 > 21% delivered during preceding 30 second epoch. |
| Non-invasive Positive Pressure Ventilation | Y/N administered during the preceding 30 second epoch. |
| Endotracheal tube | Y/N in place during the preceding 30 second epoch |
| Surfactant | Y/N administered in preceding 30 second epoch |
| LMA | Y/N in place during the preceding 30 second epoch |
| Catecholamines administered | Y/N administered in preceding 30 second epoch |
| Chest compressions | Y/N administered in preceding 30 second epoch |

diagnoses of necrotizing enterocolitis, bronchopulmonary dysplasia, bacterial infection, or intraventricular hemorrhage > Grade 1.

Exclusion criteria will include major congenital anomalies. Prematurity will not exclude participation and will analyzed as a covariate.

Goal recruitment is at least 5 normal and 5 abnormal video analyses from each site.

The overall study was reviewed by the Virginia Commonwealth University Institutional Review Board and consenting waived because of the complete anonymization of the data, such that only any consent documentation would potentially

identify a subject. Each participating site will obtain approval to participate and contribute anonymous data from their individual institutional review boards.

## Analysis

The dichotomized variables at each of the ten time points between 30 seconds and 5 minutes of life, will be counted from the normal and the abnormal videos, which should result in several hundred potential assessment tool components for evaluation. In addition, variables that permit calculation of change over time (e.g., heart rate at 150 seconds vs 30 seconds) or the effect of an earlier intervention on a component will be calculated and included in the set of potential score elements. The significance and power of each element will be calculated by Fisher's Exact Test from the proportions of each groups. Any element with a power < 0.8 will be discarded from further analysis. The individual elements will then be arranged into a hierarchy based on discriminant difference, i.e., inverse p value. Because some elements are likely to be closely related, multiple logistic regression analysis will be performed for each time point using the elements defined as significant from the proportional analysis. Those that remain both significant and independent at each time point will then be tested individually and in their various combinations for their significance between the normal and abnormal deliveries. For example, if four components make the grade for inclusion at 1 minute, analysis will be performed on each individual component, on all possible combinations of two, all combinations of three and all combinations of all four components. The ten combinations with the most significant difference will be designated as candidate newborn assessment tools to compare against the Apgar in future studies.

## Timeline

We plan to begin data abstraction from newborn resuscitation videos in the first quarter of 2025 and anticipate completing the initial 70 from the 7 sites within 6 months. The analysis and determination of the best candidate newborn assessment elements, the first step in the overall ANSWER program, should be completed by December 2025. The sites will continue to record videos for the VERIFI study, and these will be used during the first half of 2026 to test and validate a new assessment tool based on the best elements from the first step. In particular, a larger video data set from the VERIFI study will permit a further analysis of any issue with prematurity if the first phase does not have a sufficient number and range of premature deliveries.

## Discussion

For any attempt to develop or validate a newborn assessment tool or method, a short-term outcome will need to be identified. Dr. Apgar, in her original paper [1], showed that her score was lower in those babies who died in the first 28 days, but in many countries, this is an uncommon outcome and would necessitate a very large sample size. All recent attempts to modify or replace the Apgar have used different short-term outcomes [15–17]. We suggest using the assessment score to identify newborns who could benefit from enhanced observation or care after delivery, as opposed to routine care. This would include virtually all those who have conditions and consequences related to perinatal events without so many false positives as to overwhelm any systems. In the case of preterm infants, who generally go from delivery to a special or intensive care nursery, the definition would be expanded to include care not directly related to prematurity.

The second step will be to apply the Apgar score and the new assessment tools identified in the ANSWER protocol outlined above to the videos deemed Indeterminate and to the other 150 + resuscitation videos in the VERIFI registry to define their sensitivities and specificities across a broader range of deliveries. Next, the best tool sets (up to 3) will be tested for feasibility/ease of use by applying them at delivery along with the Apgar score in a single center over 6–12 months, with a goal of > 1000 deliveries. The data from this clinical trial will also be used to select the best assessment tool to test against the Apgar in large multicenter trials.

Any newborn assessment system will need to demonstrate its superiority to the Apgar score for identifying newborns with difficult or abnormal transitions to life and/or needing postnatal attention, as well as meet the basic principles outlined above. Appropriate clinical studies will include online training and testing for scorers, multiple centers to make sure that local practices do not influence outcomes, and multiple countries and regions to guard against the development of regional differences in the application of a newborn assessment program. The approach would be similar to those employed during the clinical studies to determine the best oxygen saturation goals for preterm infants, SUPPORT [18], COT [19], and BOOST II [20], where national and regional multicenter networks organized and funded the work.

This stepwise approach has several advantages. Components of a newborn assessment will reflect current practice, including fundamental interventions and will utilize video recording that can permit a reviewable analysis of a potential component. Candidate scores will be evaluated in single center tests of feasibility and utility, increasing the likelihood that the final selection for the larger international trials will be successful. Finally, by using the same definitions and protocols, data from several real-world international trials can be combined to produce sufficient power to validate a newborn scoring system for rare but critical outcomes such as long-term neurological injury.

## Author contributions

**Conceptualization:** Henry J. Rozycki, Elizabeth E. Foglia, Miheret S. Yitayew.

**Funding acquisition:** Henry J. Rozycki, Elizabeth E. Foglia, Miheret S. Yitayew.

**Methodology:** Henry J. Rozycki, Elizabeth E. Foglia, Miheret S. Yitayew, Heidi M. Herrick.

**Project administration:** Henry J. Rozycki, Heidi M. Herrick.

**Validation:** Miheret S. Yitayew, Heidi M. Herrick.

**Writing – original draft:** Henry J. Rozycki.

**Writing – review & editing:** Elizabeth E. Foglia, Miheret S. Yitayew, Heidi M. Herrick.

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
