## [Decision Letter · Decision Letter 0]

26 Feb 2025

Dear Dr. Rozycki,

Thank you for submitting your manuscript to PLOS ONE. After careful consideration, we feel that it has merit but does not fully meet PLOS ONE’s publication criteria as it currently stands. Therefore, we invite you to submit a revised version of the manuscript that addresses the points raised during the review process.

Please address the comments raised by both reviewers.

We look forward to receiving your revised manuscript.

Kind regards,

Abhishek Makkar, M.D.

Academic Editor

PLOS ONE

Journal Requirements:

 [copy in funding statement].  

3. The project described was supported by the National Center for Advancing Translational Sciences (NCATS), National Institutes of Health, through Grant Award Number 5R21TR003994 (HJR and EEF). 

Reviewers' comments:

Reviewer's Responses to Questions

**Comments to the Author**

1. Does the manuscript provide a valid rationale for the proposed study, with clearly identified and justified research questions?

Reviewer #1: Yes

Reviewer #2: Yes

2. Is the protocol technically sound and planned in a manner that will lead to a meaningful outcome and allow testing the stated hypotheses?

Reviewer #1: Yes

Reviewer #2: Yes

3. Is the methodology feasible and described in sufficient detail to allow the work to be replicable?

Reviewer #1: Yes

Reviewer #2: Yes

4. Have the authors described where all data underlying the findings will be made available when the study is complete?

Reviewer #1: Yes

Reviewer #2: Yes

5. Is the manuscript presented in an intelligible fashion and written in standard English?

Reviewer #1: Yes

Reviewer #2: No

You may also provide optional suggestions and comments to authors that they might find helpful in planning their study.

Reviewer #1: The abstract states video recordings for 25 normal and 25 abnormal resuscitations. However, in the Methods, the authors list seven participating sites with five normal and five abnormal recordings from each site, which would be a total of 35 videos each. Please clarify.

When describing the categorization of videos, it might be clearer to state normal is 4 of more criteria met, abnormal

is 2 or few criteria met, indeterminant is 3 criteria met. Also, must criteria (e) normal postnatal care be met for that delivery to be labeled "normal"? It seems criteria a-d could be met, but if the infant develops respiratory distress after 5 minutes and goes to the NICU, that might not be considered "normal".

Are there exclusion criteria for the candidate resuscitation videos that will be reviewed for the study, e.g., major

congenital anomalies? If so, please include.

The authors mention that applying the Apgar score to preterm deliveries can be challenging. Is there a plan to

ensure infants with various degrees of prematurity (eg extreme, very, moderate, late) are included in both the normal and abnormal groups?

Reference #3, the year looks incorrect.

Table 1, Newborn Status, Color is listed twice.

Reviewer #2: Congrats to the authors for working on a new tool for newborn assessment.

Few suggestions are

1. some punctuation and typographical errors in the manuscript that needs attention. for example (line 30 inter-rater reliability should be followed by a comma before the and) example line 95 has two full stops)

Also the authors also have not mentioned about how they would account for the practice difference across centers. For instance in some centers the first step in evaluating would be by auscultation where as some centers prefer EKG leads/ Dry electrodes. Are these going to be standardized across centers.

How would the 25 normal and abnormal ones be selected from the cohort is it a random selection of infants and then categorize them to normal and abnormal.

Are there any measures to control for inter-observer variability how many people would review those videos to determine normal and abnormal.

**Do you want your identity to be public for this peer review?** For information about this choice, including consent withdrawal, please see our Privacy Policy

Reviewer #1: No

Reviewer #2: No

---

## [Author Response · Author response to Decision Letter 1]

7 Mar 2025

These are the specific responses/changes to the review and comments on our above-referenced manuscript.

1. The document has been formatted to meet the referenced specifications, including author names/affiliations

2. Funding Statement has been amended in the revised cover letter to include the wording from the review.

3. See #2

4. Data availability statement now indicates “All data are in the manuscript and/or supporting information files.” as requested.

5. A paragraph regarding IRB approval and consent waiver is now included (Lines 133-137)

6. References are checked and corrected

Reviewer #1

Reviewer #1: The abstract states video recordings for 25 normal and 25 abnormal resuscitations. However, in the Methods, the authors list seven participating sites with five normal and five abnormal recordings from each site, which would be a total of 35 videos each. Please clarify.

Abstract corrected to match 35 videos in each group.

When describing the categorization of videos, it might be clearer to state normal is 4 of more criteria met, abnormal is 2 or few criteria met, indeterminant is 3 criteria met. Also, must criteria (e) normal postnatal care be met for that delivery to be labeled "normal"? It seems criteria a-d could be met, but if the infant develops respiratory distress after 5 minutes and goes to the NICU, that might not be considered "normal".

We are using all 5 criteria, including post-natal course in this phase of the score development for 2 reasons. First, one cannot know, based on the video of the first minutes of life, why a baby was admitted to a higher level of care, and there is a possibility that it was due to something that might have affected an observation in the delivery room. Including post-natal course aims keep the ‘normal population’ as clean as feasible. Second, one of the potential uses for a newborn assessment score is to identify newborns who would benefit from a higher level of post-natal care or observation, so the ‘normal’ group should not include any who do receive it.

The divisions are now defined as Normal – all 5, Abnormal - 2 or less and Indeterminant - 3 – 4, and that only Normal and Abnormal will be used. (Lines 117-119)

Are there exclusion criteria for the candidate resuscitation videos that will be reviewed for the study, e.g., major congenital anomalies? If so, please include

Sentence regarding exclusion for major congenital anomalies now added (Line 130-31)

The authors mention that applying the Apgar score to preterm deliveries can be challenging. Is there a plan to ensure infants with various degrees of prematurity (eg extreme, very, moderate, late) are included in both the normal and abnormal groups?

Sentence added to Timeline to address this: “In particular, a larger video data set from the VERIFI study will permit a further analysis of any issue with prematurity if the first phase does not have a sufficient number and range of premature deliveries.” (Lines 162-164)

Reference #3, the year looks incorrect.

Corrected

Table 1, Newborn Status, Color is listed twice.

Corrected

Reviewer #2: Congrats to the authors for working on a new tool for newborn assessment.

Few suggestions are

1. some punctuation and typographical errors in the manuscript that needs attention. for example (line 30 inter-rater reliability should be followed by a comma before the and) example line 95 has two full stops)

Corrected

Also the authors also have not mentioned about how they would account for the practice difference across centers. For instance in some centers the first step in evaluating would be by auscultation where as some centers prefer EKG leads/ Dry electrodes. Are these going to be standardized across centers.

One of the requirements for a universal newborn assessment score is that it be as universally applicable as possible so we would like to not specify how to perform in the delivery room. Observations will mainly be of the baby, and practice differences may be accounted for by the elements that measure the babies responses to interventions.

How would the 25 normal and abnormal ones be selected from the cohort is it a random selection of infants and then categorize them to normal and abnormal.

As stated, at each site, 2 people will review each video used for the VERIFI study to a) define them as normal/abnormal/indeterminate, and b) abstract the data for VRIFI and ANSWER. For this part of developing potential scoring systems, what is important is that the delivery are either normal or abnormal. Subsequent testing and validation of any score system will need to include randomization, blinding, etc. Thus, for the study protocol defined in this paper, a convenience sample of videos will be used. This now specified in Line 98

Are there any measures to control for inter-observer variability how many people would review those videos to determine normal and abnormal.

Each site has 2 reviewers for each video. Prior to starting the study, a sample video was sent to all reviewers at all sites to verify that they scored/assessed criteria equally. Furthermore, for purposed of this ANSWER PROTOCOL, reviewers are instructed to accept criteria for defining normal when they are completely sure. Combined with the objectivity of most of the criteria, (a) preductal oxygen saturation measurements meeting NRP goals without supplementation, if measured (meets objective goal or not) b) Heart rate > 100 throughout postnatal transition (meets objective goal or not) c) no apnea or respiratory distress (instructions to only accept if no question of symptoms). d) no resuscitation interventions. (direct observation of video) e) normal postnatal care (medical history) we feel this will make inter-rater reliability for defining normalcy excellent. The issue will be much more important when it comes to the individual elements in the next phases

---

## [Editor Report · Decision Letter 1]

10 Apr 2025

Assessing newborn scoring with each resuscitation (ANSWER): Protocol for identifying and testing an Apgar score for the 21st century

PONE-D-24-50763R1

Dear Dr. Rozycki,

We’re pleased to inform you that your manuscript has been judged scientifically suitable for publication and will be formally accepted for publication once it meets all outstanding technical requirements.

Kind regards,

Abhishek Makkar, M.D.

Academic Editor

PLOS ONE
---

## [Editor Report · Acceptance letter]

PONE-D-24-50763R1

PLOS ONE

Dear Dr. Rozycki,

I'm pleased to inform you that your manuscript has been deemed suitable for publication in PLOS ONE. Congratulations! Your manuscript is now being handed over to our production team.

Kind regards,

on behalf of

Dr. Abhishek Makkar

Academic Editor

PLOS ONE